# Beneficial Effects of Ursolic Acid and Its Derivatives—Focus on Potential Biochemical Mechanisms in Cardiovascular Conditions

**DOI:** 10.3390/nu13113900

**Published:** 2021-10-30

**Authors:** Jakub Erdmann, Marcin Kujaciński, Michał Wiciński

**Affiliations:** Department of Pharmacology and Therapeutics, Faculty of Medicine, Collegium Medicum in Bydgoszcz, Nicolaus Copernicus University, M. Curie 9, 85-090 Bydgoszcz, Poland; markuj94@gmail.com (M.K.); wicinski4@wp.pl (M.W.)

**Keywords:** ursolic acid, pentacyclic triterpenoid, cardiovascular disease, atherosclerosis, cardiac fibrosis, aneurysm, NF-κB pathway, reactive oxygen species

## Abstract

Ursolic acid (UA) is a natural pentacyclic triterpenoid found in a number of plants such as apples, thyme, oregano, hawthorn and others. Several in vitro and in vivo studies have presented its anti-inflammatory and anti-apoptotic properties. The inhibition of NF-κB-mediated inflammatory pathways and the increased scavenging of reactive oxygen species (ROS) in numerous ways seem to be the most beneficial effects of UA. In mice and rats, administration of UA appears to slow down the development of cardiovascular diseases (CVDs), especially atherosclerosis and cardiac fibrosis. Upregulation of endothelial-type nitric oxide synthase (eNOS) and cystathionine-λ-lyase (CSE) by UA may suggest its vasorelaxant property. Inhibition of metalloproteinases activity by UA may contribute to better outcomes in aneurysms management. UA influence on lipid and glucose metabolism remains inconsistent, and additional studies are essential to verify its efficacy. Furthermore, UA derivatives appear to have a beneficial impact on the cardiovascular system. This review aims to summarize recent findings on beneficial effects of UA that may make it a promising candidate for clinical trials for the management of CVDs.

## 1. Introduction

According to the World Health Organization, nearly 41 million deaths out of 55 million worldwide were caused by non-communicable diseases (NCDs) in 2016, representing 75% of global mortality. NCDs are not transmissible directly from one to another person, and this group of disorders includes autoimmune diseases, diabetes, cancers, cerebral strokes, respiratory diseases, cardiovascular conditions and others [1]. Of the 41 million people who died, 17.9 million deaths were caused by cardiovascular diseases (CVDs), representing 31% of all global deaths, which makes them the leading cause of mortality in the world. CVDs affect the heart or blood vessels and comprise many different types of condition (e.g., coronary heart disease, cerebrovascular disease, peripheral arterial disease and aortic disease). [2]. The most common cause of CVD is atherosclerosis, a chronic inflammatory condition that deepens over the years and remains hidden for a long time [3]. Although atherosclerosis does not tend to have any symptoms, a sudden and unexpected death may be the first symptom. However, CVDs mostly lead to disability and work inability, generating a large economic and social burden on a country [4]. The statistics show that 75% of premature CVD is avoidable [5], and prevention of these disorders may be attained by influencing modifiable factors through lifestyle changes. These elements include hypertension (most dominant), high non-HDL cholesterol, household pollution, tobacco use, unbalanced diet, low education, abdominal obesity, diabetes, low physical activity and others [6]. Among non-modifiable risk factors are age over 65 years, male gender and genetic predisposition [7]. Although the existing knowledge about prevention is well-propagated, the statistics speak for themselves. Therefore, pharmacological methods of treating cardiovascular diseases are extensively investigated. Finding a safe and effective drug, especially from a natural source, would be a breakthrough in the CVD field. Ursolic acid (UA), which belongs to the natural group known as pentacyclic triterpenoids (PTs), may have potential therapeutic or preventative effects on CVDs [8].

This review summarizes recent investigations of biological activity of UA and its derivatives in the cardiovascular system.

## 2. Pentacyclic Triterpenoids and Their Agent—Ursolic Acid

PTs constitute a structurally diverse group of natural products derived from squalene cyclization [9]. Three main classes of PTs can be distinguished: oleanane (the most common type), ursane and lupine [10]. Mostly, they exist in nature as free acids or esterified with fatty acids and as triterpenoids saponins (glycosides) linked with one or more sugar chains [11]. PTs and their derivatives are secondary plant metabolites and can be found in a variety of plant species, especially in birch bark, clove flowers, mistletoe sprouts, olive leaves and apple pomace [12]. Many medicinal herbs known from traditional Chinese medicine, such as *Ganodema lucidum*, *Boswellia serrata*, *Aster tataricus* and *Potentila discolor*, are also abundant in PTs [13]. In plants, these substances are responsible for self-defense properties against harmful life forms, communication with neighboring flora and attracting pollinators [14]. In humans, they display a wide spectrum of potential pharmaceutical properties such as anti-bacterial, anti-viral, anti-inflammatory and wound healing effects [15]. Therefore, the latest reports about PTs usage have delivered a promising approach for diabetes, vascular dysfunction and retino- and nephropathy management [10]. Moreover, PTs have received much attention in oncology because of their ability to induce apoptosis and inhibit angiogenesis, metastatis and tumor cell growth [12]. The versatility of PTs prompts scientists to further exploration and discovery of drugs for the treatment of many human diseases. Thus, one of these compounds known as ursolic acid (UA) is perceived as a future therapeutic agent that would help patients with healing cardiovascular conditions. 

UA (3β-hydroxy-urs-12-en-28-oic-acid) belongs to the ursane-type pentacyclic triterpenoids, having a C-3 hydroxyl group and C-28 carboxylic acid group. The molecular formula and weight of UA are C_30_H_48_O_3_ and 456.7 g/mol, respectively. It is acidic, has poor solubility in water and is needle-like or a crystalline solid [16]. The biosynthesis of UA is mediated by many plant enzymes through oxidization, folding and cyclization of squalene into dammarenyl. Subsequently, this molecule undergoes ring expansion and additional cyclization to create the fifth ring of UA [17]. After isolation from natural sources, UA properties have been altered by chemical reactions in the laboratory environment. It has been reported that functional groups at C-3 and C-28 are essential for the pharmacological activity of UA. Substitution at these positions allows obtaining novel UA derivatives with modified bioactivity and bioavailability [17,18]. UA concentration varies between plant species and often co-occurs with its isomer, oleanolic acid [19]. Moderately high content of UA is present in apple peel, thyme, oregano, rosemary, marjoram, sambuca, hawthorn, lavender, coffee and the wax layer of many edible fruits [17,20]. UA exerts a positive influence in various tissues and organs. In recent years, it has been reported that UA can be used as alternative medicine for the treatment and management of cancer, obesity, diabetes, brain disease, liver disease, muscle wasting (sarcopenia) and CVDs [21]. 

## 3. Ursolic Acid

### 3.1. Ursolic Acid Effect on Atherosclerosis

Atherosclerosis is a multifactorial disease with accelerated endothelial damage, which leads to an invasion of inflammatory cells into the tunica intima and adhesion of platelets. Macrophages derived from monocytes are involved in the inflammatory response and the deposition of fats, cholesterol, calcium and other substances that contribute to lipid storage and dynamic atherosclerotic plaque growth in the vascular walls. Over time, plaque hardens and arteries stiffen, which limits the flow of blood and consequently results in a hypoxic state in vital organs [22]. 

Since atherosclerosis is a chronic inflammatory state, different factors responsible for the constant recruitment and accumulation of immune cells (especially monocytes and macrophages) in plaques have been investigated. The loss of biologic activity of endothelium is associated with the increased expression of pro-inflammatory adhesion molecules, cytokines and chemotactic factors such as monocyte chemoattractant protein-1 (MCP-1), interleukin-6 (IL-6), tumor necrosis factor-α (TNF-α), intercellular adhesion molecule 1 (ICAM-1), vascular cell adhesion molecule 1 (VCAM-1), E-selectin and others [23]. Furthermore, hyperglycemia and hyperlipidemia can lead to a higher oxidative stress, which induces monocyte and macrophage dysfunction resulting in their accelerated chemotaxis. Increased chemotactic activity of abovementioned cells tightly correlates with atherosclerotic plaque size and increased recruitment into sites of vascular inflammation [16,24]. Finding an inhibiting agent of the mentioned factors appears to be logical, and it explains why the anti-atherogenic property of UA has been recently investigated intensively.

Ullevig et al. showed reduced atherosclerotic lesion formation in high-fat diet-fed diabetic mice that were LDL receptor-deficient (LDLR−/−) after 11 weeks of UA supplementation. This result was possibly obtained by the decreased release of MCP-1 from sites of vascular injury or inhibited responsiveness of monocytes/macrophages to these molecules without changing MCP-1 receptor (CCR2) surface expression. It is worth mentioning that blood glucose level was significantly reduced, whereas plasma lipid levels were not improved. In addition, UA was more effective in plaque reduction than a well-known antioxidant resveratrol (53% vs. 31% reduction in lesion size, respectively) [24]. Likewise, Nguyen et al. investigated influences of UA and its synthesized analog 23-hydroxy ursolic acid (23-OHUA) in high-fat diet-fed mice LDLR−/− for 20 weeks. Both reduced atherosclerotic plagues by protecting MAPK phosphatase 1 (MKP-1) from oxidative inactivation. Metabolic stress triggers the degradation of MPK-1, which is a counter-regulator of the MAPK pathway that stimulates adhesion and chemotactic activity of monocytes. Interestingly, analogue 23-OHUA presented a higher reduction in atherosclerotic plaque compared to UA (40% vs. 19%). Furthermore, derivative 23-OHUA reduced weight gain, whereas UA did not [25]. These results are in agreement with the statements that UA analogs can distinguish themselves from original UA by different bioactivity [17,18]. A significant reduction in atherosclerotic lesion size was also demonstrated in a study of Leng et al. in Western diet-fed mice LDLR−/− treated with UA for 11 weeks. Instead of a mechanism connected with chemotaxis, a macrophage autophagy mechanism as protective against atherosclerosis was proposed. The investigation in cell culture showed that increased mRNA expression of autophagy-related proteins (Atg5 and Atg1611) led to suppressed IL-1β secretion and enhanced promotion of cholesterol efflux from LDL-loaded macrophages to ApoA-1 through autophagy. Both effects were anti-atherogenic [26]. On the other hand, Messner and co-workers demonstrated in Western diet-fed mice that were apoE-deficient (apoE−/−) for 24 weeks that UA supplementation led to a dose-dependent increase in atherosclerotic plaque. A suggested mechanism for induction of atherogenesis was a reduction in the serum level of IL-5 (athero-protective cytokine, which stimulates removal of oxidized LDL from the circulation). However, protocol design differed from that of abovementioned studies, including differences in animal models, doses of UA and duration of supplementation. Furthermore, although UA has poor solubility in aqua, it was administered in drinking water, which may explain why UA in the serum was not detectable [27]. In contrast, in the study of Leng, UA was injected intraperitoneally, which allowed determining its plasma concentration [26]. In addition to the mice model, the same authors treated human umbilical vein endothelial cells (HUVECs) with UA to describe its exact molecular mechanism. It was found that UA induced profound cell death (apoptosis), which was initiated by DNA damage and subsequently activation of P53, leading to the release of other pro-apoptotic molecules such as BAK, cytochrome C, APAF-1 and caspases. UA-induced cell death in HUVECs indicates that UA may disrupt integrity of the vascular endothelium and promote atherosclerosis [27]. 

The impact of UA on atherosclerosis is the subject of dispute amongst many researchers; thus, some of them decided to investigate UA activity mainly on HUVECs. The study of Steikamp-Fenske et al. presented that UA upregulated endothelial-type NO synthase (eNOS), which produces nitric oxide (NO). NO is a molecule that promotes vascular relaxation and protection from thrombosis and atherogenesis. In addition, it was found that UA reduced the expression of Nox4 (NADPH oxidase subunit), which is the predominant source of reactive oxygen species (ROS) [28]. ROS impact the activity of many signaling pathways (e.g., NF-κB, MAPK, ERK1/2) that regulate gene expression. These signaling cascades may provoke endothelial dysfunction and atherosclerotic lesions progression, including their accelerated erosion and rupture [29]. For example, NF-κB is a collective name for a family of proteins: p100/p52, p105/p50, p-65 (relA), relB and c-Rel. NF-κB dimers are usually kept in the cytoplasm in a dormant state by the inhibitors of κB (IκBs). A variety of stimuli, including ROS and pro-inflammatory cytokines such as TNF-α and IL-1β, activate IκB kinases (IKKs), which phosphorylate IκBs. The phosphorylation results in the degradation of IκBs and releases NF-κB dimers, which subsequently translocate into the nucleus. In this organelle, NF-κB stimulates the expression of inflammation-associated genes, including TNF-α, IL-6, VCAM-1 and others, which creates a self-amplifying cycle [30,31]. Thus, the ability to inhibit activity of these signaling pathways may play a key role in the prevention of atherosclerosis.

The decreased production of intracellular ROS was also presented by Lin et al., who investigated the effect of UA on resistin-induced adhesion of histiocytic lymphoma cells to HUVECs. The study resulted in a suppressed adhesion between cells and downregulated expression of adhesions molecules such as VCAM-1, ICAM-1 and E-selectin due to inhibited nuclear translocation of NF-κB. It is worth mentioning that resistin, an obesity-induced adipokine, stimulates the expression of the mentioned adhesion molecules on the surface of endothelial cells, which causes accumulation of inflammatory cells in the early stage of atherosclerosis [32]. Similar results were obtained in a study of Zeller et al., in which UA application inhibited TNF-α-mediated degradation of IκB and subsequently expression of VCAM-1, ICAM-1 and E-selectin [33]. 

Interestingly, Takada et al. compared anti-atherogenic properties of two PTs: UA and oleanolic acid in HUVECs. Both of them inhibited TNF-α-induced E-selectin expression and parallel NF-κB activity. However, the UA bioactivity was attained at lower concentrations in comparison to oleanolic acid [34]. The advantage of UA over other PTs (glycyrrhetis acid, oleanolic acid) was also described by Mochizuki and co-workers. They suggested that UA structure with polar group at the 28-position is crucial for regulating the VCAM-1 expression in HUVECs [35].

Further research has been investigated in various directions. In a paper by Jiang et al., the authors described an anti-proliferation effect of UA in rat primary vascular smooth muscle cells (VSMCs). The obtained result was associated with inhibition of microRNA-21, whicht subsequently enhanced PTEN expression and then downregulation of PI3K (a molecule participating in proliferation) [36]. VSMCs were also used by Yu et al., who treated them with leptin in the presence or absence of UA. Pro-atherogenic effects of leptin such as ROS generation induction, endothelial cell activation and smooth muscle cell proliferation and migration were effectively inhibited by UA. The authors proposed a mechanism based on suppressing NF-κB and ERK1/2 signaling pathways expression and reduced ROS production. These inhibitions finally led to decreased matrix metalloproteinase-2 (MMP-2) activity. Matrix metalloproteinases are a group of proteolytic enzymes that degrade collagen and allow for smooth muscle cell migration within a vessel [37]. The above studies show that the inhibition of NF-κB activity by UA in numerous ways provides a potential protection line against a self-amplifying cycle of inflammation in developing atherosclerosis.

Low-density lipoprotein cholesterol (LDL-C) is the primary driver of atherogenesis and the key deliverer of cholesterol to the artery wall. The treatment of atherosclerosis is based on lipid-modulating therapies, which modify lipid profile by raising high-density lipoprotein cholesterol (HDL-C) and lowering LDL-C levels. Reduced LDL-C concentration is associated with lower rates of major coronary events [38]. Cholesteryl ester transfer protein (CETP) mediates the migration of cholesteryl esters from anti-atherogenic apoA-containing HDL-C to pro-atherogenic apoB particles, mainly very low-density lipoproteins (VLDLs). This process causes a decrease in HDL-C and increase in LDL-C; therefore, CETP high concentration is associated with CVDs. The in vitro studies showed that the structure of UA and OA and their derivatives allow docking into the active site of CETP protein and its inhibition. As long as their inhibitory activity is moderate, further novel UA and OA analogs with different structural scaffolds are needed to be designed to bring about the most potent efficacy [39,40].

The impact of UA on plasma lipids in mice were investigated in the previous studies, but none of them showed any improvement [24,25,26,27]. However, Li et al. presented that UA in different doses protected against elevation of total plasma cholesterol, triglyceride and LDL-C levels in high-choline diet-fed mice. UA administration also helped in maintaining balance between vasoactive components by leading to declines in levels of endothelin-1 and thromboxane A_2_ and upregulation of eNOS activity. In addition, significant reduction in aorta thickness in a dose-dependent manner was observed [41]. Another study conducted by Wang et al. presented the lipid-lowering effect of UA in Western diet-fed rabbits. UA decreased the levels of plasma cholesterol and triglyceride and the area of aortic root lesions. However, UA in combination with artesunate acid (a molecule isolated from Artemisia annua with a potential immune-modulating property) showed more potent hypolipidemic and anti-atherogenic effect than any agent alone, which indicates a strong synergistic effect. The authors also presented that UA alone decreased expression of VCAM-1 and acted as PPAR-α agonist, which may influence lipid metabolism [42].

Statin drugs as HMG-CoA reductase inhibitors allow gaining impressive reductions in LDL-C [38]. Since UA and statins may have different target points, co-administration of these medicaments seems to be reasonable. Combination of UA with simvastatin was applied in mice in the study of Li et al. It was found that this co-administration caused reduced plaque size, shrank necrotic areas and inhibited LOX-1 expression. Since LOX-1 initiates formation of atherosclerotic plaques by the uptake of oxidized LDL, its decreased activity is desirable. It is worth mentioning that no negative interactions between used therapeutic agents or adverse effects were observed. Unfortunately, UA and simvastatin were not dosed alone, and synergistic effect was not possible to assess. As in the previous studies, UA also inhibited ROS production and suppressed the activation of NF-κB, confirming its antioxidant and anti-inflammatory properties in HUVECs [43]. However, Hua et al. in their study assessed UA impact on the pharmacokinetics of rosuvastatin in rat hepatocytes. It was found that UA decreased the uptake of rosuvastatin through inhibition of OATP1B1 transporter, which suggests that a drug–drug interaction may appear [44]. 

The summary of the studies describing UA’s impact on atherosclerosis in vitro and in vivo is shown in Table 1.

### 3.2. Ursolic Acid Effect on Cardiomyocytes

Myocardial infarction (MI) is the irreversible necrosis of heart muscle due to prolonged restriction in blood supply causing a lack of oxygen. The rupture or erosion of a vulnerable atherosclerotic coronary plaque is the most common cause of MI [45]. In acute phase, ischemia induces transient metabolic and ionic alternations in the affected myocardium, while prolonged restriction causes significant pathological changes such as inflammatory cascade, apoptosis, activation of the renin–angiotensin–aldosterone system, extracellular matrix deposition, cardiac hypertrophy and ventricular remodeling [46]. As high levels of circulating catecholamines may cause MI, many scientists decided to investigate the cardioprotective effect of UA on isoproterenol-induced MI in rats. Senthil et al. presented that UA pretreatment decreased cardiomyocytes necrosis and subsequently leakage of cardiac marker enzymes (aspartate aminotransferase (AST), alanine aminotransferase (ALT), lactate dehydrogenase (LDH) and creatine phosphokinase (CPK)) to bloodstream from the damaged heart tissue. UA acted also as a free radicals and ROS scavenger, which reduced the level of myocardial lipid peroxides (thiobarbituric acid reactive substances (TBARS), lipid hydroperoxides (HPs), conjugated dienes (CDs)), which may injure blood vessels. It is also worth noting that UA blunted the myeloperoxidase (MPO) activity, which indicates that neutrophil infiltration into the injured myocardium was suppressed. In addition, the authors proposed membrane-stabilizing action of UA due to reduced peroxidation of membrane lipids, decreased ratio of cholesterol to phospholipids and increased activity of the membrane-bound phosphatases [47]. These properties of UA in cardiomyocytes were also expanded and confirmed by Radhiga et al., who performed three separated investigations. In the first study, they described reduced DNA fragmentation and subsequently blunted apoptosis by upregulation of anti-apoptotic molecules such as Bcl-2 and Bcl-xL and downregulation of pro-apoptotic proteins including Bax, caspase-3, -8, -9, cytochrome c, TNF-α and Fas. The authors also reported that UA scavenged superoxide radicals, which decreased the workload of superoxide dismutase (SOD), catalase (CAT), glutathione peroxidase (GPx), glutathione-S-transferase and glutathione reductase (GR) [48]. In the second study, macroscopic enzyme mapping of the ischemic myocardium showed a highly reduced infarct zone after UA administration. As in the previous studies [41,42], UA also increased the level of plasma HDL-C and decreased the plasma LDL-C and VLDL-C levels [49]. In the last paper, the authors presented an anti-fibrotic effect of UA through downregulated expression of MMP-2, MMP-9, collagen type I, α-smooth muscle actin (α–SMA) and transforming growth factor-β (TGF-β), which indicates that UA may be perceived as a potential therapeutic agent in prevention of cardiac fibrosis. In hypoxia condition caused by MI, mitochondrial tricarboxylic acid cycle and respiratory chain enzymes activities are blunted, which results in vulnerability to degradation and low production of ATP molecules. UA treatment showed increased activities of abovementioned enzymes, possibly by protecting “SH” group of dehydrogenases from free radicals attack, which allowed to maintain oxygen consumption. In addition, UA decreased activities of lysosomal glycohydrolases and cathepsins, which may stimulate infiltration of inflammatory cells at the location of infarction and further cardiac muscle tissue damage [50]. A study of Al-Taweel et al. presented the effect of *Nepeta deflersiana* extract, in which UA was one of nine secondary metabolites isolated from the plant. In addition to results similar to the previous studies, suppressed NF-κB activity in cardiomyocytes and decreased levels of TNF-α, IL-6 and IL-10 were described [51]. 

The restoration of blood flow to the ischemic myocardium may be associated with further myocardial injury. Apoptosis, autophagy and oxidative stress are involved in this harmful process. However, an increased level of uncoupling protein 2 (UCP2), a cationic carrier protein, appears to possess a protective effect in hypoxia–reoxygenation injury. Thus, Chen et al. tested six natural compounds in rats cells under ischemia–reperfusion conditions. Among them, UA exhibited the most effective impact on UCP2 expression, which was possibly attained by inhibition of p38 signaling pathway. Furthermore, the authors noted an anti-apoptotic property based on depleted caspase-3 activity and antioxidative effect through decreased ROS production, MDA content and increased SOD activity [52].

Cardiomyopathy is a medical term for a heterogeneous group of progressive diseases with structural and functional dysfunctions of the heart, usually with improper ventricular hypertrophy or dilatation. There are four major types, but dilated cardiomyopathy is the most common and mainly caused by coronary artery diseases or hypertension in adults. However, the causes of cardiomyopathies are varied, including environmental factors and genetic predispositions [53]. The development of myocardial fibrosis in cardiomyopathy is characterized mainly by myofibroblasts overactivity, an increased collagen type I deposition in the extracellular matrix, pro-fibrotic actions of transforming growth factor β (TGF- β), oxidative stress, inflammatory cytokines, endothelin-1 and the renin–angiotensin–aldosterone system [54]. For instance, a chronic abuse of alcohol leads to alcoholic cardiomyopathy. Thereby, Saravanan et al. decided to check UA’s cardioprotective effect in rats that were persistently administered rats. After the induction of toxicity by ethanol for 30 days, treatment group underwent UA therapy for another 30 days. The cardioprotective effect of UA was demonstrated by a decreased level of myocardial lipid peroxides, increased activity of free radical scavenging enzymes and an elevated level of non-enzymic antioxidants such as reduced glutathione, ascorbic acid and α-tocopherol [55]. Another mechanism was described by Yang et al., who induced dysfunction of mouse cardiac myocytes by heat stress. It was found that UA reduced mortality through inducing the expression of anti-apoptotic protein Mcl-1, which inhibited Puma pathway and subsequent apoptosis [56]. Doxorubicin belongs to anthracycline class of drug and is commonly used in cancer chemotherapy. The cardiac cell apoptosis activated by ROS is a well-known side effect of doxorubicin. Mu et al. applied this medicament to mice in order to asses UA role in cardiac protection. The results showed that UA improved left ventricular fractional shortening (LVFS) and left ventricular ejection fraction (LVEF) of the heart. These effects were possibly gained by the increased expression of eNOS and the inhibited expression of Nox4 [57], which was also previously described by Steinkamp-Fenske et al., but in human endothelial cells [28]. Improved structural and hemodynamic parameters were also presented by Wang et al. As diabetes mellitus is one of the greatest public health emergencies, they decided to administer UA for 8 weeks in rats with diabetic cardiomyopathy. The cardiac improvement was attained by antioxidative effect of UA and depleted expressions of TNF-α, MCP-1 and TGF-β_1_ in the heart, while the level of MMP-2 almost normalized compared to the control group. In addition, this was the second study in which blood glucose level was decreased [24,58]. It is worth noting that TGF-β_1_ is one of the best-characterized fibrogenic growth factors, and its presence induces collagen deposition, inhibition of interstitial collagenases and extracellular matrix protein synthesis. TGF-β_1_ activation is dependent on other molecules such as MMP-2, MMP-9 and TSP-1 [54]. Thus, depletion of these activating molecules’ content or direct inhibition of TGF-β_1_ by UA may imply its strong anti-fibrotic property. Dong et al. tried to expand the biochemical aspect of UA and TGF-β_1_. They implied that UA competed with TGF-β_1_ binding to its receptor, which occurred along with inhibition of microRNA-21 and MAPK/ERK1/2 signaling pathways [59]. 

Pulmonary arterial hypertension (PAH) is a progressive disease characterized by high blood pressure in the pulmonary arteries and right ventricle remodeling, ultimately leading to chronic heart failure. As the current treatment of this disorder is far away from adequate, Gao et al. administered UA on PAH-induced right ventricle dysfunction in rats. Ameliorated cardiac hypertrophy, cardiac fibrosis and improved hemodynamic parameters of right ventricle were observed after 4 weeks of UA treatment. The authors suggested that the underlying mechanism was based on the increased expression of PPARα and CPT1b, crucial genes regulating fatty acid metabolism and inhibition of pro-apoptotic factor Bax [60]. 

The above studies show that treatment of cardiomyocytes with UA influences numerous cell signaling enzymes and simultaneously protects them against hypoxic and fibrotic conditions. Therefore, further research is needed to determine more detailed biochemical connections between UA activity and signaling pathways. The summary of the studies describing UA’s impact on cardiomyocytes in vitro and in vivo is shown in Table 2. 

### 3.3. Ursolic Acid Effect on Vasodilatation

Hypertension affects every developed society in the world, and by 2030, the prevalence is estimated to reach 37.3% in the US. Despite the presence of antihypertensive drugs with various mechanisms of action, 47.5% of treated patients still do not achieve therapy goals [61]. Incorrect blood pressure numbers induce oxidative stress in the vascular wall, endothelium dysfunction and aggravation of atherosclerosis [23]. Hypertension may be a result of a reduction in the concentration of endothelial relaxation factors, an increased reactivity to contractive stimulus in comparison to healthy individuals or changes occurring in the vascular endothelium itself [62]. Currently conducted investigations show that vasorelaxation of vessels is mainly dependent on NO and endothelium-derived hyperpolarizing factors (e.g., potassium ion, myo-endothelial gap junctions, epoxyeicosatrienoic acids). Furthermore, hydrogen sulfide (H_2_S) is produced by cystathionine-λ-lyase (CSE) and cystathionine-β-synthase (CBS). H_2_S is a multi-tasking factor that plays a key role in vascular homeostasis by maintaining its integrity, relaxation and stimulatory effect on the NO signaling pathway [63,64]. The above-mentioned studies have already shown that UA reduced the level of endothelin-1 (endothelium contracting factor) [41] and increased eNOS activity, which produces NO [28,41,51,57] and therefore may imply its vasorelaxant property. This feature was firstly detailed by Aguirre-Crespo and co-workers, who incubated rat aortic rings with UA and by turns with another vasorelaxant or vasoconstrictor agents. They found that UA-mediated relaxation was endothelium-dependent, possibly by boosting eNOS and NO release, which activated vascular smooth muscle soluble guanylate cyclase (sGC), a signal transduction enzyme that converts GTP to cGMP [65]. The UA-mediated upregulation of eNOS and activation of NO/cGMP pathway was verified by Luna-Vazguez et al. They also presented another mechanism of UA vasorelaxation based on increased activity of CSE and H_2_S release that activates K_ATP_ channels located in vascular smooth muscle cells. Additionally, in silico study supported the hypothesis that UA attaches directly to an allosteric binding site in eNOS and CSE, which stabilizes the quaternary structure of the active sites [66]. The in vivo investigation on spontaneously hypertensive Wistar rats confirmed vasorelaxant property of UA. They were treated with a single intragastric dose of UA, which led to a significant decrease in systolic and diastolic blood pressure (SBP, DBP) without modifying heart rate. It is worth noting that captopril was more potent in reducing SBP than UA, but lowering DBP was similar [67]. However, a chronic administration of UA and its effect on blood pressure after a longer period of time, including adverse effects, were not assessed.

### 3.4. Ursolic Acid Effect on Aneurysm

The abdominal aortic aneurysm (AAA) is a localized enlargement of the abdominal aorta that affects mainly male elderly people. Not only male sex is an independent risk factor, but also smoking and high blood pressure. Asymptomatic AAA is mainly managed conservatively, but there are two main types of surgery offered to patients: open surgery or minimally invasive endovascular repair. However, considering the pathogenesis of aneurysms, pharmacological prevention or treatment should be investigated to find a medical therapy which may be effective at reducing the growth rate and rupture rate [68]. Pathophysiology of AAA is complex but can be characterized by inflammation of the aortic wall, oxidative stress, apoptosis of smooth muscle cells, modification of the extracellular matrix, breakdown of elastin and atherosclerosis [69,70]. Based on current research, it is known that some of these processes are dependent on matrix metalloproteinases such as MMP-2 and MMP-9, whose activities may be attenuated by inhibition of STAT3 and NF-κB pathways [71,72]. The disintegrin and metalloproteinase 17 (ADAM17), also called tumor necrosis factor-α converting enzyme (TACE), is responsible for the release of TNF-α in a soluble form that binds to TNF receptor 1 (TNFR1), resulting in stimulation of the NF-κB pathway [73,74]. In the previous studies, it was shown that UA was able to inhibit NF-κB in numerous ways [32,33,34,35,37,43] and MMPs activity [37,50,62], which may suggest its validity of usage in the treatment of aneurysms. 

A study by Zhai et al. on male mice apoE (−/−) with elevated cholesterol levels showed that treatment with UA not only reduced the incidence of angiotensin II-induced AAA but also decreased mortality. In addition, UA increased continuity of aorta by maintaining stability of elastin and reducing the intensity of VSMCs proliferation. In the same study, using immunohistochemistry allowed the authors to find that UA reduced the production of MMP-2, MMP-9, ADAM17 and phosphor-STAT3 (pSTAT3), which may explain the phenotype changes that occurred [73].

Another mechanism was suggested by Huang C et al., who investigated UA influence on VSMCs of aortic aneurysm. The proliferation and migration of VSMCs were inhibited by UA administration through the miR-16/PTEN/PI3K/AKT/mTOR signaling pathway [75].

## 4. Ursolic Acid Derivatives and Their Effects on the Cardiovascular System

Among the ursane-type triterpenoids, UA seems to be the most studied member because of its broad spectrum activity. This review has already presented that UA displays promising therapeutic potential in cardiovascular conditions. However, ursane-type triterpenoids comprise a number of bioactive agents, such as asiatic acid, corosolic acid, 23-hydroxy ursolic acid, pomolic acid, uvaol and others [25,76]. Their similar structure to UA may suggest that these agents should also exert positive effects on the cardiovascular system. Thus, discovering and investigating UA’s natural or synthetic derivatives may bring promising candidates for further studies. In discussion below, UA derivatives’ activity will be limited to the cardiovascular system.

### 4.1. Asiatic Acid

Centella asiatica, commonly known as Gotu Kola, is widely grown in tropical and subtropical countries. One of the constituents contained in this plant is asiatic acid (AA) with its multiple pharmacological activities in the cardiovascular system [8]. Kamble et al. injected doxorubicin in Wistar rats to induce multi-organ toxicity. They found that pretreatment with AA in a dose-dependent manner (5, 10, 20 mg/kg per os) ameliorated oxidative stress in liver, kidney and heart, which resulted in decreased activity of damage biomarkers and better histological outcome of these organs. The authors suggested that AA afforded protection against doxorubicin toxicity mainly by increased expression of the NrF2 protein, which modulates cells’ response to ROS [77]. Prevention of doxorubicin-induced cardiotoxicity by AA was also obtained by Hu et al. in mice. These authors presented that AA activated the protein kinase B (AKT) signaling pathway, which is tightly connected with NrF2 expression [78]. Further studies have focused on the protective effect of AA against cardiac hypertrophy. A study of Ma et al. was conducted using mice after aortic banding, a procedure that increases left ventricular systolic pressure. It was found that AA orally given (10 or 30 mg/kg) for 7 weeks suppressed hypertrophic response caused by pressure overload. AA restored antioxidant capacity of AMP-activated protein kinase α (AMPKα) and inhibited mTOR pathway and extracellular signal-regulated kinase (ERK), which are key players in the process of cardiac hypertrophy [79]. Likewise, Li et al. used angiotensin II peptide hormone to induce cardiac hypertrophy in rats. Therapy of rats with AA decreased cardiac hypertrophy markers and heart weight, possibly due to suppression of miR-126 and activation of the PI3K/AKT signaling pathway [80]. What is more, Wu et al. applied an in vitro model based on human cardiomyocytes to evaluate the positive effect of AA under hypoxic conditions. They showed that AA enhanced the survival of these cells, and the molecular mechanism behind this change was dependent on the regulation of miR-1290/HIF3A/HIF-1 α axis [81].

### 4.2. Corosolic Acid

The leaves of Eriobotrya japonica and Lagerstroemia speciosa, commonly called “Japanese loquat” and “Banaba”, respectively, are abundant in corosolic acid (CA) [82]. Its beneficial activity for cardiovascular health was first described by Chen et al. in Western diet-fed mice apoE−/−. It was found that CA reduced atherosclerotic lesion formation and decreased serum glucose and triglycerides levels. In line with in vitro study, CA depleted the expression of MCP-1 and inhibited chemotactic activity of monocytes along with downregulation of NF-κB [83]. Another in vivo study associated with CA protective activity on vessel endothelial homeostasis was performed by Li et al., who described a complex biochemical mechanism. In response to oxidative stress, CA inhibited mitochondrial fission by induction of Drp1 phosphorylation (Ser637). That manifested its action in suppression of NOX2 oxidase signaling, which prevented NLRP3 inflammasome activation and ultimately cell apoptosis in endothelium [84]. Further studies have focused on the protective effect of CA in artificially induced MI in mice. Sahu et al. presented that Lagerstroemia speciose leaves extract containing 1% CA prevented vacuolization, loss of myofibrils and myocardial necrosis. From a biochemical standpoint, it was attained by restoration of antioxidant enzymes such as Nrf2/Ho-1 and decreased lipid peroxidation [85]. Similar results were replicated by Wang et al., but the underlying mechanism was different. It was found that CA ameliorated TGF-β_1_ expression and reversed activity of AMPKα [86]. The same authors confirmed a heart-protective feature of CO in mice with developed cardiac hypertrophy due to aorta banding [87]. Interestingly, Torres-Ortiz et al. incubated rat aortic rings with the extract from the flowers of Crataegus gracilior. They showed that the extract possessed a concentration-dependent vasodilator property, and the authors suggested that the main agent responsible for that activity was CO. However, contributions of other flavonoids and triterpenes could not be excluded [88]. 

### 4.3. 23-Hydroxy Ursolic Acid

23-OHUA can be obtained from the stem bark of Cussonia bancoensis. Its structure is close to asiatic acid, and the only difference is the lack of the hydroxyl group at position C-2 [89]. In this review, 23-OHUA has been already presented as a more effective molecule in reducing atherosclerotic plaque and weight gain in high-fat diet-fed mice LDLR−/− compared to UA. Its anti-atherogenic effect was possibly gained by protecting MKP-1 from oxidative inactivation [25]. Another study conducted by the same authors confirmed the prevention of weight gain in high-calorie diet-fed mice supplemented with either 0.05% or 0.2% 23-OHUA. It is worth mentioning that 23-OHUA reduced leptin and cholesterol levels and improved glucose tolerance but did not affect insulin sensitivity, which may suggest that 23-OHUA acts through novel anti-diabetic pathways. These positive effects of 23-OHUA, as in the previous investigation, were explained by its ability to preserve MKP-1 activity in monocytes, which protected these cells from chronic nutrient stress-induced priming and transformation into a pro-inflammatory phenotype [90].

### 4.4. Pomolic Acid

Pomolic acid (PA) isolated from Licania pittieri has also been demonstrated to possess a protective effect on the cardiovascular system. Estrada et al. incubated rat aortic rings with PA and vasoconstrictor agents. They found that PA possibly led to the direct activation of endothelial purinergic P2 receptors and the NO-cGMP signaling pathway, which induced a vasodilator effect [91]. The same authors revealed that PA was capable of inhibiting ADP-induced aggregation of human platelets in vitro by acting as a competitive antagonist of P2Y_12_ receptor [92]. A study of López et al. presented that PA may also exert a positive effect on isolated cardiomyocytes from rats. PA preserved sarcoplasmic reticulum from Ca^2+^ exhaustion, which enhanced the rate of cardiomyocytes relaxation (positive lusitropic effect) [93]. Further insights into the molecular mechanisms of action of PA may expand its list of cardiovascular properties.

## 5. Ursolic Acid and Human Subjects

### 5.1. Randomized Controlled Trials

Ramírez-Rodríguez conducted a double-blind study on 24 patients (30–60 years-old) with metabolic syndrome who were randomized to supplement orally with 150 mg of UA or placebo once a day for 12 weeks. In the group receiving the intervention, 50% of them noted transient remission of metabolic syndrome (reduced body weight, body mass index, waist circumference and fasting glucose concentrations), whereas in the control group, no one achieved such remission [94]. On the other hand, a double-blind study of Cione et al. on 26 postmenopausal women (61 ± 7 years) that lasted 8 weeks and was connected with an exercise intervention program presented no therapeutic effect of UA (450 mg/day) supplementation compared to the placebo group [95]. What is more, Lobo et al. assessed the impact of UA on muscle strength and body mass gain in active men (18-35 years-old) on a high-protein diet for 8 weeks. They stated that an isolated UA dosage of 400 mg/d did not provide any beneficial effects compared to the placebo group [96]. Interestingly, Church et al. demonstrated on nine physically active men that consuming 3 g of UA before resistance exercise does not affect the Akt/mTORC1 signaling pathway or serum IGF-1 and insulin hormones concentrations. Importantly, peak plasma UA concentration of 103 ng/mL occurred after 2 h following oral administration of 3 g of UA (35.93 mg/kg^−1^). The authors compared the obtained result to the pharmacokinetic data that presented a peak plasma UA concentration of 3404 ng/mL at 4 h after administering the liposomal formulation of UA (98 mg/m^−2^) via intravenous injection [97]. This observation is in agreement with the physico-chemical property of UA that is poor solubility in water, which may interfere in its absorption in the gastrointestinal tract [16,20,98]. This characteristic may also explain inconsistent results in the abovementioned studies. Based on the above research, it is worth mentioning that oral administration of UA in humans was well-tolerated, and adverse effects did not occur or were not statistically significant [94,95,96,97]. 

### 5.2. Clinical Trials, Phase I

Clinical trials are essential steps in evaluating the safety and efficacy of potential therapeutic agents. Extensive preclinical studies have clearly shown that UA is a promising drug candidate for the treatment of many contemporary diseases and allowed moving UA into clinical stage. UA is presently under phase I, and by now, only results of three studies have been published [99,100,101]. In the People’s Republic of China, a total of 108 healthy volunteers and patients with advanced solid tumors were enrolled. Due to potential uncertainties associated with the bioavailability, UA was administered in the liposomal formulation via intravenous injections. Importantly, all clinical trials presented that UA had a linear pharmacokinetic profile, and accumulation was not observed despite repeated doses of UA. The intravenous infusion was well tolerated, and adverse effects varied from mild to moderate. The most reported alarming events were abdominal distention, nausea, diarrhea and skin rash. Only one participant experienced a grade 3 adverse event in the form of elevated liver enzymes with diarrhea. Further phase II clinical trials should be performed to provide a deeper understanding of UA activity and its full clinical potential.

## 6. Conclusions

The information presented in this article allows for considering UA and its derivatives as potential drugs in the treatment of CVDs. Concluding from the preclinical in vitro and in vivo studies, UA seems to be a multi-tasking substance that regulates many transcription factors, protein kinases and other molecular agents. The inhibition of the pro-inflammatory NF-κB pathway, diminishment of free radical creation and an anti-apoptotic property may prove to be helpful in slowing down atherosclerosis and cardiac fibrosis developments. UA impacts aneurysm growth and possesses a potential vasorelaxant property that needs to be established in the future studies. However, influence on lipid and glucose metabolism remains inconsistent. It is evident that the therapeutic activity of UA can be improved through modifications at different positions. Fresh novel derivatives may need to be synthesized to increase their solubility, bioavailability and potency. This study shows that UA from natural products and its natural or laboratory-synthesized derivatives may represent important cardio- and vasoprotective medicaments and may be promising therapeutical agents in the management of CVDs. However, additional studies are necessary to verify the efficacy of UA in conditions specified in the review, particularly in humans.

## Figures and Tables

**Table 1 nutrients-13-03900-t001:** Ursolic acid—effects and proposed mechanisms of its activity in managing atherosclerosis.

Author	Subject of Study	Potential Mechanism of Action
Ullevig et al., 2011 [24]	High-fat diet-fed diabetic mice LDLR−/−	↓ atherosclerotic plaque size, ↓ blood glucose level Decreased release of MCP-1 from sites of vascular injury or inhibited responsiveness of monocytes/macrophages to these molecules
Nguyen et al., 2018 [25]	High-fat diet-fed mice LDLR−/−	↓ atherosclerotic plaque size, ↓ weight gain protection of MAPK phosphatase 1 (MKP-1) from oxidative inactivation
Leng et al., 2016 [26]	LPS-stimulated cell culture RAW264.7	Increased mRNA expression of autophagy-related proteins (Atg5 and Atg1611) in macrophages suppresses IL-1β secretion and enhances promotion of cholesterol efflux from LDL-loaded macrophages to ApoA-1 through autophagy
	Western diet-fed mice LDLR−/−	↓ atherosclerotic plaque size, ↓ serum level of IL-1β
Messner et al., 2011 [27]	HUVECs	Pro-atherogenic property by induction of apoptosis Causation of DNA damage activates P53, which allows creating BAK dimers that mediate the release of pro-apoptotic factors (cytochrome c, APAF-1), subsequently leading to caspases-3 and -9 activation and cell death
	Western diet-fed mice apoE−/−	↑ atherosclerotic plaque size, ↓ serum level of IL-5
Steinkamp-Fenske et al., 2007 [28]	EA.hy 926 endothelial cells and HUVECs	Upregulation of eNOS, which produces NO Inhibited expression of Nox4, which is the predominant source of ROS
Lin et al., 2016 [32]	Resistin-stimulated lymphoma cells (U937) and HUVECs	Blunted generation of ROS and nuclear translocation of NF-κB, which suppresses adhesion between cells through decreased expression of VCAM-1, ICAM-1 and E-selectin
Zeller et al., 2012 [33]	TNF-α-stimulated HUVECs	Decreased degradation of IκBs, which inhibits expression of VCAM-1, ICAM-1 and E-selectin
	Wistar rats after autologous vein grafting	Decreased expression of VCAM-1 Reduced intimal hyperplasia through induction of non-inflammatory-type smooth muscle cells death
Takada et al., 2010 [34]	TNF-α-stimulated HUVECs	Inhibited NF-κB activity and decreased expression of E-selectin
Mochizuki et al., 2019 [35]	TNF-α-stimulated HUVECs	Inhibited degradation of IκBs, which reduces expression of VCAM-1
Jiang et al., 2016 [36]	10% fetal-bovine-serum-stimulated rat VSMCs	Inhibition of microRNA-21, which enhances PTEN expression, and subsequently downregulation of PI3K expression
Yu et al., 2017 [37]	Leptin-stimulated rat VSMCs	Suppressed NF-κB and ERK1/2 signaling pathways, ROS generation, which subsequently leads to reduced MMP-2 activity
Chang et al., 2017 [39]	CETP enzymatic inhibition assay	Inhibition of CETP
Chen et al., 2020 [40]	CETP enzymatic inhibition assay	Inhibition of CETP
Dongyu Li et al., 2016 [41]	High-choline diet-fed mice	↓ total plasma cholesterol, ↓ triglyceride, ↓ LDL-C, ↓ endothelin-1, ↓ thromboxane A_2_ ↓ aorta thickness Upregulation of eNOS, which produces NO
Wang et al., 2013 [42]	Western diet-fed rabbit	↓ total plasma cholesterol, ↓ triglyceride, ↓ area of aortic root lesions Suppressed expression of VCAM-1 PPAR-α agonist activity
Qiu Li et al., 2018 [43]	LPS-stimulated HUVECs	Reduced endothelial LOX-1 expression in mRNA and proteins levels through decreased ROS generation and inhibited nuclear translocation of p65 NF-κB
	Atherogenic diet-fed mice apoE−/−	↓ atherosclerotic plaque size through inhibited expression of LOX-1 in the aorta (ursolic acid in combination with simvastatin)
Hua et al., 2014 [44]	Rat hepatocytes	Decreased uptake of rosuvastatin through inhibition of OATP1B1 transporter

↑—increase; ↓—reduction; LDLR−/−—low-density lipoprotein receptor-deficient; MCP-1—monocyte chemoattractant protein-1; LPS—lipopolysaccharide; MKP-1—MAPK phosphatase 1; IL-1β—interleukin-1β; HUVECs—human umbilical vein endothelial cells; apoE−/−—apolipoprotein E-deficient; IL-5—interleukin-5; eNOS—endothelial-type NO synthase; NO—nitric oxide; Nox4—NADPH oxidase 4; ROS—reactive oxygen species; VCAM-1—vascular cell adhesion molecule-1; ICAM-1—intercellular cell adhesion molecule-1; TNF-α—tumor necrosis factor-α; IκBs—inhibitors of κB; VSMCs—vascular smooth muscle cells; MMP-2—matrix metalloproteinase-2; CETP—cholesteryl ester transfer protein.

**Table 2 nutrients-13-03900-t002:** Ursolic acid—effects and proposed mechanisms of its activity in cardiomyocytes.

Author	Subject of Study	Potential Mechanism of Action
Senthil et al., 2007 [47]	Isoproterenol-stimulated Wistar rats	↓ cardiomiocytes necrosis and ↓ leakage of cardiac marker enzymes (AST, ALT, LDH, CPK) Acting as a scavenger of free radicals and ROS, which reduced the level of myocardial lipid peroxides (TBARS, HPs, CDs) Inhibited MPO Membrane-stabilizing property due to decreasing the ratio of cholesterol to phospholipids and increasing activity of the membrane-bound phosphatases (Na^+^K^+^ATPase, Ca^2^^+^ATPase and Mg^2^^+^ATPase)
Radhiga et al., 2012 [48]	Isoproterenol-stimulated Wistar rats	↓ cardiomiocytes necrosis and subsequently leakage of cardiac marker enzymes (CK-MB, cTnT, cTnI) Acting as a scavenger of free radicals and ROS, which reduced the level of myocardial lipid peroxides (TBARS, HPs, CDs) and the workload of enzymatic antioxidants, which maintained their activity (SOD, CAT, GPx, GST and GR) ↓ DNA fragmentation and subsequently blunted apoptosis by upregulation of anti-apoptotic molecules such as Bcl-2 and Bcl-xL and downregulation of pro-apoptotic proteins such as Bax, caspase-3, -8, -9, cytochrome c, TNF-α and Fas
Radhiga et al., 2012 [49]	Isoproterenol-stimulated Wistar rats	↓ cardiomiocytes necrosis and ↓ leakage of cardiac marker enzymes (CK, CK-MB, LDH) ↑ HDL-C, ↓ LDL-C, ↓ VLDL-C Antioxidative property due to acting as a scavenger of free radicals and ROS, which reduced DNA damage
Radhiga et al., 2019 [50]	Isoproterenol-stimulated Wistar rats	↓ MMP-2, ↓ MMP-9, ↓ collagen type I, ↓ α-SMA, ↓ TGF-β Increased activities of tricarboxylic acid cycle and respiratory chain enzymes possibly through protection of “SH” group of dehydrogenases, which maintains oxygen consumption Reduced activities of lysosomal glycohydrolases and cathepsins
Al-Taweel et al., 2017 [51]	Isoproterenol-stimulated Wistar rats	↓ cardiomiocytes necrosis and ↓ leakage of cardiac marker enzymes (AST, CK-MB, LDH) ↓ TNF-α, IL-6, IL-10 Increased levels of SOD, CAT and NP-SH Inhibited MPO activity Upregulation of eNOS, which produces NO Blunted apoptosis by upregulation of anti-apoptotic molecules such as Bcl-2 and downregulation of pro-apoptotic proteins including Bax and caspase-3 Suppressed NF-κB activity
Chen et al., 2018 [52]	Rat H9c2 cells under ischemia-reperfusion injury	Increased level of UCP2 through inhibition of p38 signaling pathway ↓ caspase-3 ↓ ROS, MDA and increased SOD activity ↑ NO
Saravanan et al., 2006 [55]	Ethanol-treated Wistar rats	↓ cardiac marker enzymes (CPK, LDH) Acting as a scavenger of free radicals and ROS, which reduced the level of myocardial lipid peroxides (TBARS, LOOH, CDs) and the workload of enzymatic antioxidants, which maintained their activity (SOD, CAT, GPx, GST), and increased non-enzymic antioxidants (reduced glutathione, ascorbic acid and α-tocopherol)
Yang et al., 2014 [56]	Heat-stress-treated ICR mice	Decreased level of MDA and increased level of reduced glutathione in the heart Anti-apoptotic property through increasing level of Mcl-1, which inhibited PUMA pathway
Mu et al., 2019 [57]	Doxorubicin-treated ICR mice	Upregulation of eNOS, which produces NO Inhibited expression of Nox4, which is the predominant source of ROS
Wang et al., 2018 [58]	Diabetic Sprague–Dawley rats	↓ cardiac marker enzymes (CK, LDH) ↓ blood glucose level Increased activity of SOD and decreased level of MDA ↓ TNF-α, ↓ MCP-1, ↓ TGF-β_1_ in the heart, while the level of MMP-2 almost normalized compared to the control group
Dong et al., 2015 [59]	TGF-β_1_-treated cardiac fibroblast from neonatal Kunming mice hearts	Inhibition of microRNA-21 and MAPK/ERK1/2, which occurred along with downregulation of TGF-β_1_ ↓ α-SMA
	Kunming mice after transverse aortic constriction	Anti-fibrotic property associated with inhibition of microRNA-21 and MAPK/ERK1/2
Gao et al., 2020 [60]	Neonatal rat ventricular myocytes from neonatal Sprague–Dawley rats	Inhibited expression of pro-apoptotic Bax factor
	Sprague–Dawley rats with pulmonary arterial hypertension	Decreased mRNA levels of ANP and BNP Decreased mRNA levels of type I and type III procollagen (COL1A1, COL3A1) and TGF-β_1_

↑—increase; ↓—reduction; AST—aspartate aminotransferase; ALT—alanine aminotransferase; LDH—lactate dehydrogenase; CPK—creatine phosphokinase; ROS—reactive oxygen species; TBARS—thiobarbituric acid reactive substances; HPs—lipid hydroperoxides; CDs—conjugated dienes; MPO—myeloperoxidase; CK-MB—creatine kinase MB; cTnT—cardiac troponin T; cTnI—cardiac troponin I; SOD—superoxide dismutase’ CAT—catalase; GPx—glutathione peroxidase; GST—glutathione-S-Transferase; GR—glutathione reductase; TNF-α—tumor necrosis factor-α; HDL-C—high-density lipoprotein cholesterol; LDL-C—low-density lipoprotein cholesterol; VLDL—very low-density lipoprotein; MMP-2—matrix metalloproteinase-2; MMP-9—matrix metalloproteinase-9; TGF-β—transforming growth β; α-SMA—α-smooth muscle actin; NP-SH—non-protein sulphydydryl; eNOS—endothelial-type NO synthase; NO—nitric oxide; UCP-2—uncoupling protein 2; MDA—malondialdehyde; Nox4—NADPH oxidase 4.

## Data Availability

Excluded.

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
