# Peer review of "Beneficial Effects of Ursolic Acid and Its Derivatives—Focus on Potential Biochemical Mechanisms in Cardiovascular Conditions"

_nutrients, 2021, doi:10.3390/nu13113900_

Round 1

Reviewer 1 Report

Dear Authors:

Your manuscript title is “Beneficial Effects of Ursolic Acid and Its Derivatives - Focus on Potential Biochemical Mechanisms in Cardiovascular Conditions.” This manuscript is writing about ursolic acid effects on atherosclerosis, including cardiomyocytes, vasodilatation, and aneurysm. I think this review manuscript wants to let people know that ursolic acid is a therapeutic agent. As the following file is my recommendation:

Author Response

Dear Sir / Madam,

                I am writing to you regarding manuscript entitled “Beneficial Effects of Ursolic Acid and Its Derivatives - Focus on Potential Biochemical Mechanisms in Cardiovascular Conditions. I am sending you the manuscript with the adjustments made.

 My recommendations:

  1. Abstract:

That is ok. That is better.

Introduction

  1. Pentacyclic Triterpenoids and Their Agent – Ursolic Acid
  2. Ursolic Acid as a Therapeutic Agent for the Management of Atherosclerosis

2-1. Effect on Cardiomyocytes

2-2. Effect on Vasodilatation

2-3. Effect on Aneurysm

  1. Ursolic Acid and Its Derivatives
  2. ??? You can write some about this topic. Need more two subsections.

References

More references above to 100.

Plag Scan 6.5%. This is fine. I can accept this range.

Many words spell wrong. I think that is ok.

                Firstly I would like to thank you for your relevant suggestions and positive feedback. I was advised to do minor and major corrections to my manuscript. In order to make the manuscript more transparent and to improve the construct I have edited and rephrased it as you recomennded. I have also expanded references above to 100. Hence I have added 2 sections (in the first one I have presented Ursolic acid derivatives and their effects on the cardiovascular system, whereas in the second one Ursolic acid and human subjects have been described). Moreover I have transformed 3 sections into 3 subsections. All these adjustments have been made with the aim of improving the clarity of the text and putting the information in order. I have also corrected some mistakes in English.

Ursolic acid from fruit or herbal medicine?

I have mentioned from which fruit or herbal medicines ursolic acid can be obtained. However, I have not expanded this issue because in the most studies, the authors only mention that ursolic acid was purchased in a special institution or a store and the source of ursolic acid is not specified.

Ursolic Acid and Derivatives Exhibit Anti-atherosclerotic Activity?

In the most in vitro and in vivo studies ursolic acid exhibits anti-atherosclerotic activity. However, ursolic acids derivatives cannot be firmly specified as anti-athersoclerotic due to the lack or small number of performed investigations.

Respectfully yours,

Jakub Erdmann

Reviewer 2 Report

This manuscript (review article) entitled "Beneficial Effects of Ursolic Acid and Its Derivatives - Focus on Potential Biochemical Mechanisms in Cardiovascular Conditions" by Erdmann et al, has discussed the potential role of Ursolic Acid and Its Derivatives in Cardiovascular pathophysiological conditions. Unfortunately, this manuscript is a collection of detailed information. It is also very complex and hard to follow. It would be better if the authors could keep it simple and informative.

Author Response

Unfortunately, this manuscript is a collection of detailed information. It is also very complex and hard to follow. It would be better if the authors could keep it simple and informative.

Dear Sir / Madam,

                I am writing to you regarding manuscript entitled “Beneficial Effects of Ursolic Acid and Its Derivatives - Focus on Potential Biochemical Mechanisms in Cardiovascular Conditions. I am sending you the manuscript with the adjustments made.

                Firstly I would like to thank you for your relevant suggestions. In order to make the manuscript more transparent and to improve the construct I have edited and rephrased it. Hence I have added 2 sections (in the first one I have presented Ursolic acid derivatives and their effects on the cardiovascular system, whereas in the second one ursolic acid and human subjects have been described). Moreover I have transformed 3 sections into 3 subsections. All these adjustments have been made with the aim of improving the clarity of the text and putting the information in order.

                Since this is the scientific dissertation on a typically biochemical topic, it is characterised by a high level of details and a large amount of strictly specialist vocabulary. As is well known, biochemistry at the molecular level is very complicated. If I had described it briefly and easier – i.e. by simplifying it - the quality of this scientific work would decrease and might even be unacceptable to the scientific community. At the same time, many molecular factors are being analysed and a description of all of them is necessary. The tables originally included are intended to simplify the interpretation of the results of changes at the molecular level.

Respectfully yours,

Jakub Erdmann

Round 2

Reviewer 1 Report

Dear Authors:

Thank you.

Reviewer 2 Report

The manuscript has improved a little bit after revision.